# Hydrogen Production by Formic Acid Decomposition over Ca Promoted Ni/SiO_2_ Catalysts: Effect of the Calcium Content

**DOI:** 10.3390/nano9111516

**Published:** 2019-10-25

**Authors:** B. Faroldi, M. A. Paviotti, M. Camino-Manjarrés, S. González-Carrazán, C. López-Olmos, I. Rodríguez-Ramos

**Affiliations:** 1Instituto de Catálisis y Petroleoquímica, CSIC, C/Marie Curie 2, 28049 Madrid, Spain; cristina.lopez.olmos@csic.es; 2Instituto de Investigaciones en Catálisis y Petroquímica (INCAPE-CONICET), Facultad de Ingeniería Química, Universidad Nacional del Litoral, Santiago del Estero 2829, Santa Fe 3000, Argentina; apaviotti@fiq.unl.edu.ar; 3Departamento de Química Inorgánica, Facultad de Ciencias Químicas, Universidad de Salamanca, 37008 Salamanca, Spain; mcamino@usal.es (M.C.-M.); silviag@usal.es (S.G.-C.)

**Keywords:** hydrogen production, formic acid decomposition, nickel catalyst, calcium oxide promoter, silica support

## Abstract

Formic acid, a major product of biomass processing, is regarded as a potential liquid carrier for hydrogen storage and delivery. The catalytic dehydrogenation of FA to generate hydrogen using heterogeneous catalysts is of great interest. Ni based catalysts supported on silica were synthesized by incipient wet impregnation. The effect of doping with an alkaline earth metal (calcium) was studied, and the solids were tested in the formic acid decomposition reaction to produce hydrogen. The catalysts were characterized by X-ray diffraction (XRD), X-ray photoelectron spectroscopy (XPS), temperature-programmed reduction (TPR), Fourier transform infrared spectroscopy (FTIR), transmission electron microscopy (TEM), and programmed temperature surface reaction (TPSR). The catalyst doped with 19.3 wt.% of Ca showed 100% conversion of formic acid at 160 °C, with a 92% of selectivity to hydrogen. In addition, all the tested materials were promising for their application, since they showed catalytic behaviors (conversion and selectivity to hydrogen) comparable to those of noble metals reported in the literature.

## 1. Introduction

Hydrogen (H_2_) has significant advantages as an energy vector compared to petroleum or other conventional fossil fuels, although currently there are problems associated with its production, storage, and transportation that must be solved [1]. One possible solution for mobile applications, such as fuel cells, is to produce H_2_ in situ by a reaction such as the reforming of methanol, although these reactions have been associated with the generation of CO_2_, a greenhouse gas. There would be a significant advantage if H_2_ was produced from a chemical product derived from biomass since, in it, the CO_2_ formed in parallel must be considered a product of a carbon-neutral balance process. Formic acid is a chemical substance of relatively low specific volume and has limited uses, including its application as an antibacterial and antifungal agent. It can be produced by chemical methods such as the hydrolysis of methyl formate, but it is also obtained in equimolar proportions, together with levulinic acid, by hydrolysis of cellulose raw materials derived from biomass. Currently, with the increased interest in the production of levulinic acid and other valuable chemicals from biomass, it is important to develop processes to use the derived formic acid, since, otherwise, it constitutes a waste material [1]. In this direction, the interest in the use of the decomposition reaction of formic acid to produce H_2_ has increased remarkably. Therefore, the challenge is to produce pure H_2_, with minimal CO content, at the lowest possible temperature. This demand can be achieved through the careful choice of the catalyst and the reaction conditions, so a lot of research is currently being done in this direction. 

The production of H_2_ from formic acid using heterogeneous catalysts has been studied in liquid [2] and vapor phases [3], but, in most cases, formulations based on noble metals, such as Rh, Pt, Ru, Au, Ag, and Pd that are supported on C, Al_2_O_3_, and SiO_2_ have been investigated [1,2,3]. For the vapor-phase reaction, Solymosi et al. [3] found the following order of activity on a set of carbon-supported noble metals: Ir > Pt > Rh > Pd > Ru. They reported that the decomposition of formic acid started at and above 77 °C on all catalysts, and that decomposition was complete at 200–250 °C. On the other hand, in the case of non-noble metals, the activities of supported Ni catalysts have been measured, proving to be active at relatively higher temperatures (>220 °C) than noble metal catalysts [4].

Ni catalysts with different support matrices show adequate activity and selectivity for H_2_ production in various processes [5,6,7,8]. SiO_2_ stands out among the supports used due to its high surface area and low acidity [5,6,8]. Alkaline-earth metal oxides act as structural promoters by increasing the dispersion of the active phase and stabilizing the dispersed metallic phase against sintering [9,10,11,12]. Also, these additives act as chemical promoters by influencing the acid–base properties of support [13,14,15] or the electron density of dispersed metal crystallites [16,17]. In particular, the use of CaO as a promoter has exerted a positive effect on the increase in the interaction of the Ni with the support and the resistance of the Ni catalysts to the sintering [18,19,20].

In this work, Ni catalysts supported on a SiO_2_ matrix are used in the decomposition reaction of formic acid in the vapor phase. The effect of doping the support with different loadings of calcium is studied. The catalysts were characterized by X-ray diffraction (XRD), X-ray photoelectron spectroscopy (XPS), temperature-programmed reduction (TPR), Fourier transform infrared spectroscopy (FTIR), transmission electron microscopy (TEM), and programmed temperature surface reaction (TPSR).

## 2. Materials and Methods 

### 2.1. Catalysts Preparation

Ni catalysts were supported on commercial SiO_2_ AEROSIL 200 (S_BET_ = 200 m^2^/g, Degussa, previously calcined at 900 °C) or on Ca-SiO_2_ solids. The binary supports were prepared by incipient wetness impregnation of SiO_2_ with Ca(NO_3_)_2_·6H_2_O (Panreac Química SLU, Castellar del Vallès, Spain). Different Ca loadings were used (3.4, 6.8, and 19.3 wt.%). The Ca–SiO_2_ supports were maintained at room temperature for 12 h and then dried in an oven at 90 °C overnight. The solids thus obtained were finally calcined in flowing air, at 550 °C for 6 h. Samples are denoted as Ca(X)–SiO_2_, where X stands for the nominal Ca content in wt.%.

The Ni metal was incorporated by incipient wetness impregnation with a concentration of 5 wt.%, using Ni(NO_3_)_2_·6H_2_O (Alfa Aesar, Thermo Fisher Scientific, UK) as the precursor. These samples were subjected to a drying process in equal conditions to those of the binary support.

### 2.2. Sample Characterization

The surface area of the material was measured by BET analysis of the N_2_ adsorption isotherms collected at −196 °C (ASAP 2020 Micromeritics Instrument Corp., Norcross, GA, USA), with pretreatment at 200 °C for 2 h. The crystalline phases of the samples were examined by X-ray diffraction (XRD), using an X’Pert Pro PANalytical B.V., Almelo, The Netherlands. The TPR experiments were carried out in a conventional fixed-bed flow reactor, and the effluent gases were continuously monitored by online mass spectrometry (Pfeiffer/Balzers Quadstar GmbH, Asslar, Germany, QMI422 QME125); the samples were heated up in a 5% H_2_/Ar stream with a rate of 10 °C/min up to 800 °C. The XPS measurements were carried out using a multi-technique system (SPECS GmbH, Berlin, Germany) equipped with a dual Mg/Al X-ray source and a hemispherical PHOIBOS 150 analyzer. The catalysts were analyzed after two reduction treatments under H_2_ atmosphere, first at 400 °C for 1 h in a tubular quartz reactor and then at 400 °C for 15 min in the load-lock XPS chamber. Transmission electron microscopy (TEM) images of the reduced catalysts were acquired using a JEOL, Ltd., Tokyo, Japan, JEM 2100F field emission gun electron microscope equipped with an energy dispersive X-ray (EDX) detector. The fresh samples were reduced at 400 °C for 1 h in a pure H_2_ stream, while used samples were measured without any treatment. The particle size was determined by counting 300 particles. The temperature-programmed surface reaction (TPSR) measurements were carried out in conventional dynamic vacuum equipment coupled to a quadrupole mass spectrometer (RGA-200, SRS Inc., Sunnyvale, CA, USA). The catalysts were reduced before experiments in hydrogen flow at 400 °C and were degassed in a high vacuum at the same temperature. The adsorption was then carried out using a 40 Torr pulse of HCOOH at 40 °C. Once the gas phase was evacuated, the desorption step was carried out at a programmed temperature, and the gases released were analyzed with a mass spectrometer.

### 2.3. Catalytic Test

The catalytic activity measurements for the formic acid decomposition in the vapor phase were carried out in a conventional fixed-bed flow reactor. The catalysts were pretreated in H_2_ flux at 400 °C for 1 h and then cooled in N_2_ flux at the reaction temperature. A mixture of formic acid diluted with N_2_ was fed to the reactor using a saturator–condenser at 15 °C (HCOOH concentration equal to 6%, with a flow of 25 mL·min^−1^). The reactants and products were analyzed by gas chromatography with a Carboxen 1000 column and a TCD detector.

## 3. Results

### 3.1. Catalysts Characterization

Figure 1 shows the diffractograms obtained for the Ni/SiO_2_ and Ni over the binary supports after reduction in hydrogen at 400 °C for 1 h. From the XRD patterns, the characteristic diffraction broad peak centered on 2θ = 23° confirmed the amorphous nature of silica in Ni/SiO_2_ sample. No reflections from CaO species were observed in the diffraction patterns obtained for the Ca(X)-SiO_2_ supported catalysts.

The XRD patterns of Ni/SiO_2_ and Ni/Ca(19.3)-SiO_2_ catalysts exhibit broad peaks which could be assigned to the Ni species. The XRD peaks at 2θ = 44.5°, 51.9°, and 76.4° indicate the presence of metallic nickel (JCPDS 04-0850), although small broad peaks at 2θ = 37.2°, 43.3° and 62.9° reveal that there is also oxidized nickel phase (JCPDS 47–1049) [21].

The FTIR spectra of the reduced Ni/SiO_2_ and Ni/Ca(19.3)-SiO_2_ catalysts were analyzed (Figure 2). FTIR spectra show a broad band at 3528–3596 cm^−1^, which corresponds to the stretching vibration mode of the O-H bond from the silanol group (Si-OH). The band at 1050–1080 cm^−1^ is assigned to the asymmetric stretching vibration of the siloxane bonds (Si-O-Si). The network Si-O-Si symmetric bond stretching vibrations are found at 620–900 cm^−1^, whereas the network O-Si-O bending vibration modes are observed at 469–481 cm^−1^. It is noted that the bands decrease in intensity as the content of Ca wt.% increases [20]. For the Ca promoted sample, with 19.3 wt.% of Ca, the broad band at 1458 cm^−1^, associated with the band at 876 cm^−1^, is assigned to asymmetric C-O stretch and out-of-plane deformation, respectively, of monodentate carbonate species on the CaO phase [22].

The XPS analysis of the reduced catalysts was carried out to study the presence of different surface Ni species (Figure 3). Both samples presented the Si 2s peak at 154.8 eV and the O 1s simple signal centered at 533 eV, corresponding mainly to the photoemission of the oxygen atoms presented on the siliceous support [23]. The Ca 2p_3/2_ core level spectrum from the reduced sample shows binding energy of 348.2 eV [24].

Figure 3 shows a difference in the Ni 2p_3/2_ spectra; the Ni catalysts exhibited a peak at 853.2 eV associated with reduced Ni species and another contribution related with octahedral Ni^2+^ clusters at 856.3 and 857.1 eV for Ni/SiO_2_ and Ni/Ca(19.3)-SiO_2_, respectively [25]. This difference in the binding energy values could be related to a different interaction between the Ni and the support. In both catalysts, a reduced nickel fraction was observed under the treatment conditions carried out prior to the catalytic tests. This Ni^0^/Ni^2+^ surface ratio was higher for the Ni/SiO_2_ catalyst. From the deconvolution of the spectra, the surface concentration of Ni^0^ was estimated with respect to the total of surface Ni species, resulting in 70% and 13% for Ni/SiO_2_ and Ni/Ca(19.3)-SiO_2_, respectively.

The reducibility of the supported nickel catalysts was studied by temperature-programmed reduction (TPR). TPR is a powerful tool for the study of the reduction behavior of oxidized phase, as NiO, and obtainment of the strength of the oxide-support interaction. Figure 4 shows the TPR profiles of the catalysts, where two main reduction peaks can be observed. Peaks in the 200–300 °C range that are attributed to the reduction of superficial oxygen [11] are not detected in these solids. Peaks above 300 °C represent reductions in Ni(II) species with different interactions with the support. Peaks between 300 °C and 600 °C can be attributed to Ni(II) species with low support interaction, as NiO. Due to high mobility, this Ni(II) phase can be easily reduced and shows a low reduction temperature. Peaks above 600 °C refer to Ni(II) with moderate/strong support interaction [26]. Reduction peaks at higher temperatures appear as Ca is added, which suggests this addition increases the interaction of Ni(II) with the support. The reduction temperature increased with the Ca loading. These results are consistent with those observed by XPS experiments.

The TEM images of the undoped and doped Ca(19.3) materials are shown in Figure 5. It can be seen that the nickel particles are evenly distributed over the support. To estimate the average size, 300 particles were measured. Values of 5.1 and 4.8 nm for the undoped and doped catalyst, respectively, were obtained. Thus, the doping with Ca did not modify the average size of the particles, but did slightly modify the particle size distribution (see histograms in Figure 5).

Figure 6 shows the images obtained in the STEM mode and the EDX mapping of nickel (green), calcium (red), and silicon (white) revealed that Ni and CaO particles are evenly distributed on SiO_2_. In addition, the EDX images showed that Ni particles coincided in space with CaO phase supporting the metal-support interaction revealed by the XPS and TPR experiments.

### 3.2. Catalytic Performance in Fixed-Bed Reactor

Ni catalysts supported on SiO_2_ and on Ca-SiO_2_ were used in the decomposition reaction of formic acid to produce hydrogen. The catalytic activity of all materials was evaluated by operating the reactor in the experiments with a mass/flow ratio (W/F) equal to 5 × 10^−5^ g·h·mL^−1^. The decomposition of formic acid can give the following as products:
HCOOH(g) → H_2_ + CO_2_(1)
HCOOH(g) → H_2_O + CO(2)
The values of reaction temperature for which the materials reach a 50% or 100% conversion of formic acid and its H_2_ selectivity are compared in Table 1.

It can be observed that the Ni/SiO_2_ catalyst reached the 50% conversion at a temperature of 148 °C and 100% at 180 °C, while the selectivity was 91% and 87%, respectively (Table 1). In the catalysts supported on binary systems, the selectivity was higher in all cases. It is important to note that the catalyst with the highest Ca content (19.3 wt.%) reached 100% conversion at 160 °C, this being 20 °C lower than the undoped one.

Liu et al. [27] reported the study of the effect of different temperature pretreatments and atmospheres on the catalytic behavior of Ni catalysts for the dry reforming of methane. They observed that materials treated in He compared with those treated in H_2_ achieved better yields. During the pretreatment with He, a small fraction of Ni particles was reduced. However, a short period of exposure to reactants was sufficient to achieve the formation of metallic Ni nanoparticles that are particularly active under reaction conditions [27]. This phenomenon could explain the high activity of the Ni/Ca(19.3)-SiO_2_ catalyst, even though a low proportion of surface metallic species was observed after the reduction treatment.

Figure 7 shows the conversion of formic acid as a function of the reaction temperature for the series of Ni catalysts. The light-off curves were made following the same procedure in all the samples. After the reduction of the catalytic material, it was cooled in N_2_ flux to 60 °C, and then the reaction mixture was fed with a concentration of HCOOH of 6% in N_2_. After the curve measured from 0% to 100% (1st evaluation—Figure 7), the temperature was lowered to leave it in isothermal conditions and to measure the stability of the samples (Figure 8). After 16 h of reaction, the temperature was lowered and the complete curve was again measured from 0% to 100% (2nd evaluation in Figure 7) of the light off curve. The behavior throughout the conversion range shows the same tendency observed in Table 1. The catalysts were relatively stable under the conditions tested, although it can be observed that the points corresponding to the 2nd evaluation are below those obtained in the 1st, probably due to a restructuring of the material at the temperature reached (160–180 °C) and with conversion values close to 100%. In the Ca(19.3)-SiO_2_ catalyst, this behavior is less marked.

The doping of K in Pd catalysts supported over SiO_2_, Al_2_O_3_, and activated carbon was previously reported [28]. These authors observed a significant effect of improvement in the catalytic behavior of noble metal for the formic acid decomposition. As a reaction mechanism, they proposed, as a first step, the formation of a phase containing liquid formic acid condensed in the pores of the catalyst; this phase provides a reservoir for the formation of formate ions with the participation of K^+^ ions that later decompose to form CO_2_ and H_2_. In our materials, since the support is a nonporous material, condensation of formic acid is not likely to occur in pores; however, formates could form in the alkaline earth oxyhydroxide phase in the doped catalysts, with these species being the reaction intermediates.

### 3.3. Study of the Adsorbed Species Under Reaction Conditions: Temperature Programmed Surface Reaction and FTIR Experiments

Temperature-programmed surface reaction (TPSR) experiments were carried out to try to understand the differences in the catalytic performance. The catalysts were reduced before experiments in hydrogen flow at 400 °C and were degassed in high vacuum at the same temperature. The adsorption was then performed using a pulse of 40 Torr of HCOOH at 40 °C. Once the gas phase was evacuated, the evolution of the masses desorbed as a function of temperature was followed by mass spectroscopy. The TPSR experiments for the Ni catalysts are shown in Figure 9. Among the detected gases are the evolution of H_2_, CO_2_, CO, H_2_O, and HCOOH (m/z = 2, 44, 28, 18, and 29, respectively). At lower temperature (<100 °C) the desorption of the unreacted HCOOH is observed, and, above 80 °C, the decomposition process begins to produce H_2_ and CO_2_ and minority CO and H_2_O. It can be observed that the undoped catalyst exhibits lower adsorption of HCOOH and, subsequently, lower production of H_2_ and CO_2_. In all the samples, the molar ratio CO_2_/H_2_ produced was equimolar, as corresponds to the decomposition of formic acid (Table 1). These values were calculated by integrating the H_2_ and CO_2_ signals and taking into account the relative calibration of these gases.

There are two regions marked on the profiles: the first part corresponds to the decomposition of formic acid (up to 180 °C), and the second part corresponds to the decomposition of surface or mass species (formates, bicarbonates, and carbonates). The second part is closely related to the basic component of the catalyst. These samples doped with Ca present the H_2_ and CO_2_ desorption at a higher temperature. This could indicate greater stability of the species, for example, formate or bicarbonate species, which store hydrogen.

### 3.4. Characterization of Used Catalysts

The FTIR spectra of the used Ni catalysts were analyzed (Figure 10). Figure 10a shows FTIR spectrum of used Ni/SiO_2_ catalysts; the reduced one was included for comparison. It can be clearly observed that the spectra are identical before and after the catalytic test, and the signals described above are present. Figure 10b shows the spectrum of used Ni/Ca(19.3)-SiO_2_ catalysts; the reduced one was included for comparison.

For all the samples, the fingerprints of SiO_2_ at 475, 805, and 1115 cm^−1^ related to the Si-O-Si, Si-OH, and Si-O bonds, respectively, are observed. The characteristic bands of surface monodentate carbonate species at 1458 and 876 cm^−1^ are not present in the used Ni/Ca(19.3)-SiO_2_ catalyst. In addition to SiO_2_ bands, characteristic bands of formate species at 2877 and 1377 cm^−1^ are observed in the calcium doped solids used in reaction (Figure 10b,c). These bands are very weak in the spectrum of Ca(3.4)-SiO_2_, but they are more intense at high X values (Ca(19.3)-SiO_2_). The signal centered at 1645 cm^−1^ and the shoulder at 1240 cm^−1^ revealed the presence of asymmetric C-O stretching a C-O-H bending modes, respectively, of bicarbonate species [22,29]. For all the Ca-SiO_2_ supported catalysts, the presence of formate and bicarbonate species in the catalysts after the reaction was confirmed by FTIR (Figure 10c) and were consistent with TPSR experiments.

Figure 11 shows the diffractograms obtained for the used Ni/SiO_2_ and Ni/Ca-SiO_2_ catalysts. The characteristic diffraction broad peak centered on 2θ = 23° confirmed the amorphous nature of the SiO_2_ support. After reaction experiments, the diffraction patterns obtained for the Ca(X)-SiO_2_ supported catalysts exhibit reflections from CaO species centered at 15.8°, 26.5°, and 30.7° [11]. The used Ni/Ca(19.3)-SiO_2_ catalyst, as well as the reduced one, present broad peaks assigned to Ni species, indicating the presence of both metallic and oxidizes Ni particles. This result is consistent with those observed through TPR and XPS experiments. For Ni/SiO_2_ catalyst, the peaks at 44.3°, 51.7°, and 76.2° are assigned to metallic Ni particles. Comparison with the fresh reduced samples (Figure 1) reveals that, in the case of the Ni/SiO_2_ sample, the reflections corresponding to Ni^0^ became sharper, which suggest that, in the absence of the Ca promoter, nickel is affected by the reaction conditions.

The TEM images of the undoped and doped Ca materials are shown in Figure 12. It can be observed that the nickel particles are evenly distributed over the support.

The estimated particle size using around 300 particles was 8.9 and 4.8 nm for the undoped and doped catalyst, respectively. The histogram of the Ni/SiO_2_ particles was modified during the catalytic test, and the distribution and average particle size are doubled with respect to those of the reduced sample. However, the doping with Ca modified the interaction of the metal with the support and the Ni particles remained stable during the catalytic test (see histograms in Figure 12). These findings are in agreement with the XRD results discussed above.

The stability in the average size of the nickel particles in the used Ni/Ca(19.3)-SiO_2_ catalyst could explain the difference in the behavior of this material with respect to the others presenting an equal performance in both catalytic tests (1st and 2nd evaluation in Figure 7). In addition, it can justify the highest stability of this catalyst during the long-term experiments of this reaction (Figure 8).

Figure 13 shows the images obtained in the STEM mode and the EDX mapping of nickel (green), calcium (red), and silicon (white) revealed that Ni and CaO particles are evenly distributed on SiO_2_ and coincide in occupying the same space on the support.

## 4. Conclusions

Ni catalysts supported on SiO_2_ and on Ca-SiO_2_ were synthesized. These materials were employed in the formic acid decomposition reaction to produce hydrogen. On the catalytic system, calcium species act as structural promoters by stabilizing the dispersed metallic phase against sintering. In addition, this additive also acts as a chemical promoter by influencing the acid–base properties of support and the interaction metal-support.

The XRD patterns for Ni/SiO_2_ and Ni/Ca(19.3)-SiO_2_ catalysts indicated the presence of metallic and oxidized nickel particles after the reduction step at 400 °C. The reduction temperature increased with the Ca loading. These results are consistent with those of the XPS and XRD experiments. The average size of Ni particles was measured for the two samples, undoped and doped catalyst, being 5.1 and 4.8 nm, respectively. The incorporation of Ca did not modify the average size of the particles, but did slightly modify the particle size distribution.

The Ni/SiO_2_ catalyst reached 50% of formic acid conversion at a temperature of 148 °C and 100% at 180 °C, while the selectivity was 91% and 87%, respectively. For the catalysts supported on binary systems, the selectivity was higher in all cases. It is important to note that the catalyst with the highest Ca content (19.3 wt.%) reached 100% conversion at 160 °C, this being 20 °C lower than that of the undoped one. The doping with Ca modified the interaction of the metal with the support and the Ni particles remained stable during the catalytic test. However, for Ni/SiO_2_ catalyst the distribution and average particle size are doubled during reaction with respect to those of the reduced sample. The stability in the average size of the nickel particles in the used Ni/Ca(19.3)-SiO_2_ catalyst could explain the difference in the behavior of this material to the others. Moreover, this catalyst was relatively stable under the reaction conditions used, presenting an equal performance in two sequential catalytic tests.

The TPRS experiments reveal that, at lower temperatures (<100 °C), the desorption of the unreacted HCOOH is observed, and above 80 °C, the decomposition process begins to produce H_2_ and CO_2_ and minority CO and H_2_O. It can be observed that the undoped catalyst exhibits lower adsorption of HCOOH and subsequent lower production of H_2_ and CO_2_. In the Ca-SiO_2_ supported catalysts, the presence of formate and bicarbonate species in the catalysts after the reaction was confirmed by FTIR and were consistent with TPSR experiments.

## Figures and Tables

**Figure 1 nanomaterials-09-01516-f001:**
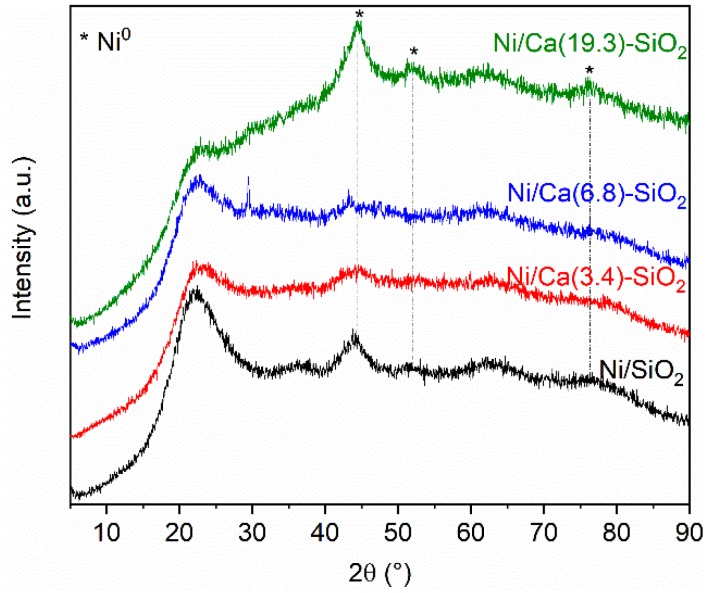
X-ray diffractograms of reduced Ni/SiO_2_ and Ni/Ca(X)-SiO_2_ catalysts.

**Figure 2 nanomaterials-09-01516-f002:**
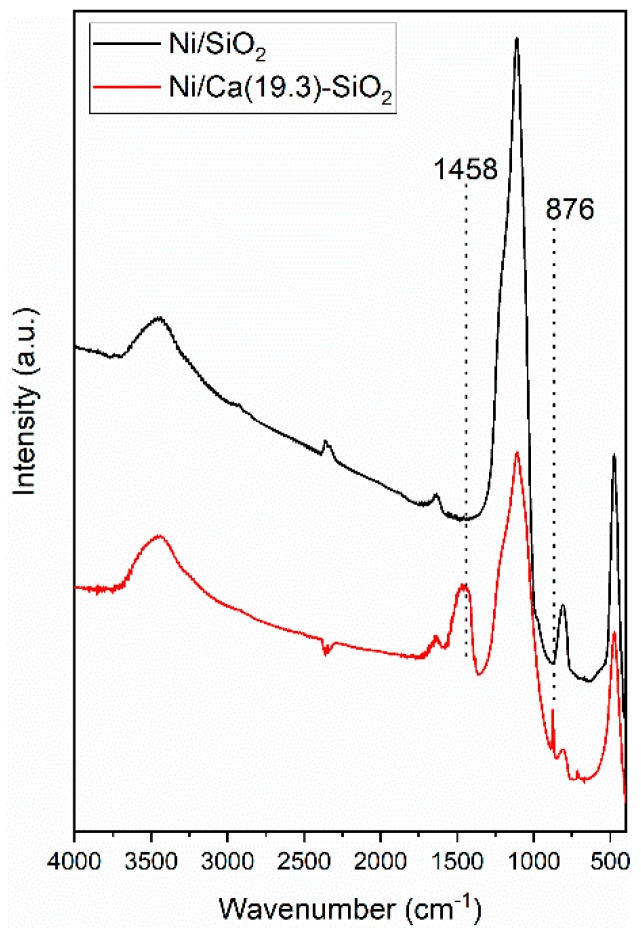
Fourier transform infrared spectroscopy (FTIR) spectra of reduced Ni/SiO_2_ and Ni/Ca(19.3)-SiO_2_ catalysts.

**Figure 3 nanomaterials-09-01516-f003:**
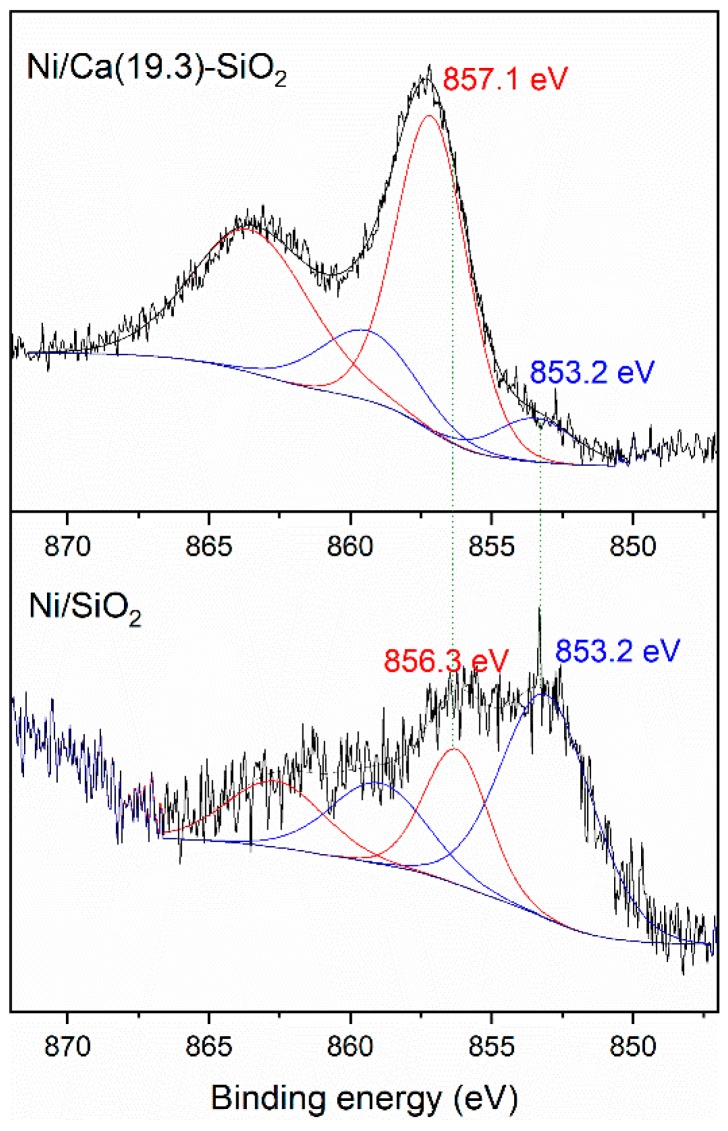
X-ray photoelectron spectroscopy (XPS) spectra of Ni 2p_3/2_ region of reduced Ni/SiO_2_ and Ni/Ca(19.3)-SiO_2_ catalysts.

**Figure 4 nanomaterials-09-01516-f004:**
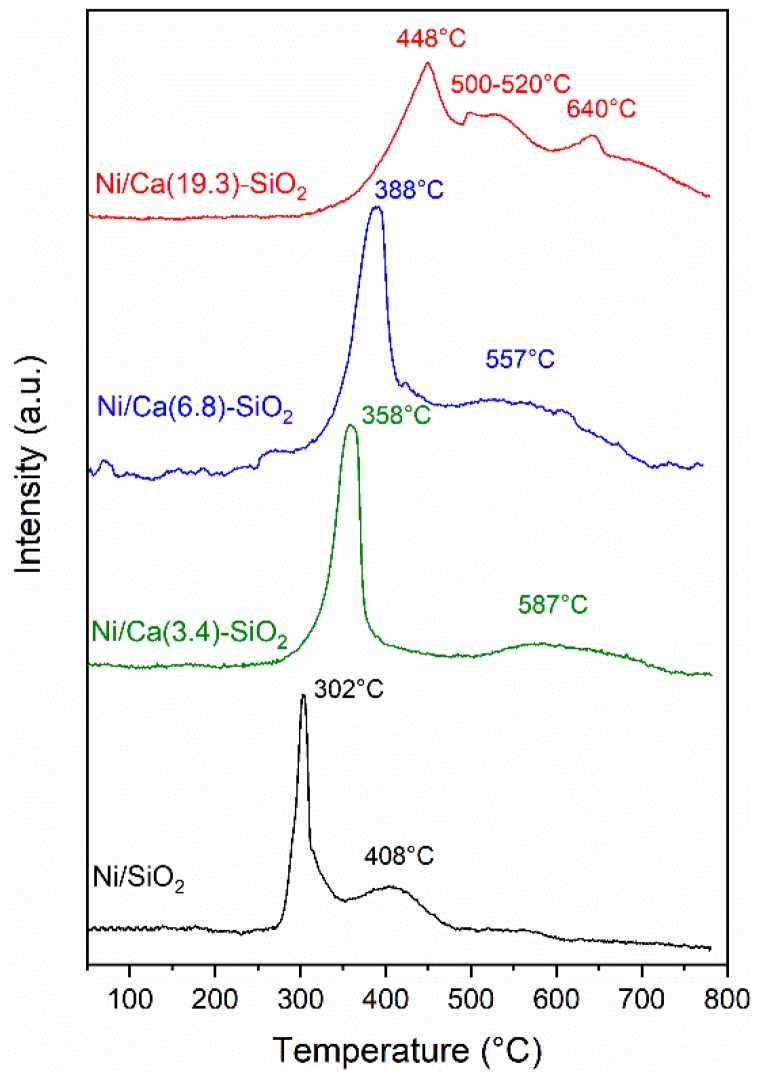
Temperature-programmed reduction (TPR) profiles of Ni/SiO_2_ and Ni/Ca(X)-SiO_2_.

**Figure 5 nanomaterials-09-01516-f005:**
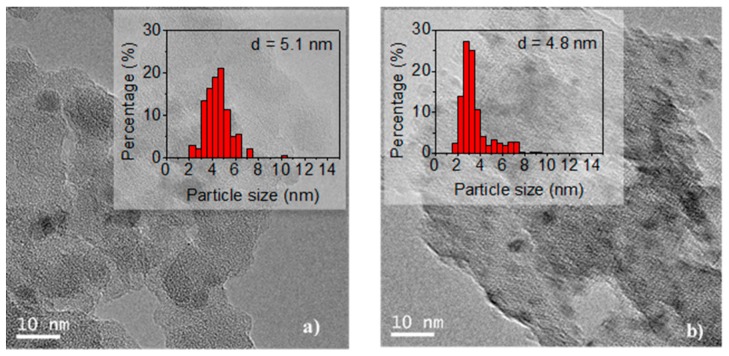
Transmission electron microscopy (TEM) images of reduced samples: (**a**) Ni/SiO_2_ and (**b**) Ni/Ca(19.3)-SiO_2_; the histograms were included.

**Figure 6 nanomaterials-09-01516-f006:**
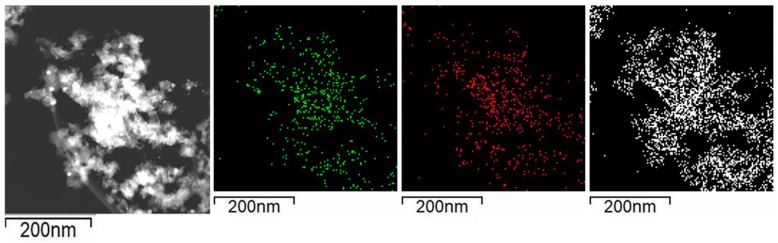
Selected area for the EDX mapping in reduced Ni/Ca(19.3)-SiO_2_; mapping of nickel (green), calcium (red), and silicon (white).

**Figure 7 nanomaterials-09-01516-f007:**
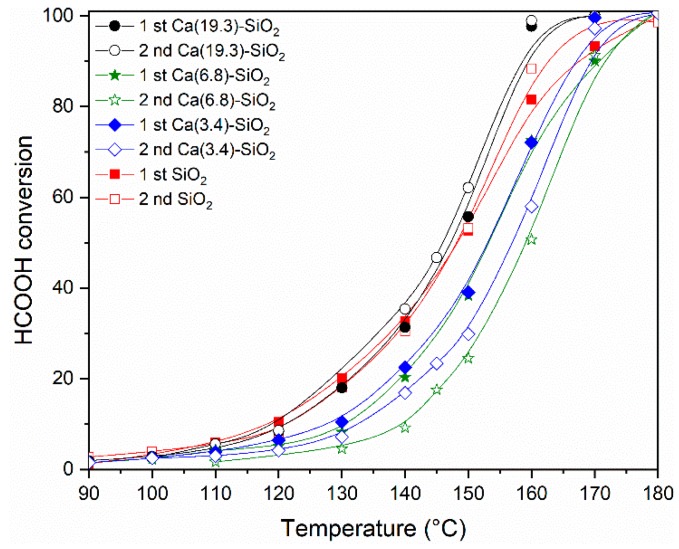
Catalytic activity of Ni solids after reduction at 400 °C at different reaction temperatures. The HCOOH conversions are plotted as function of the reaction temperature (W/F = 5 × 10^−5^ g·h·mL^−1^; feed composition: 6% HCOOH/N_2_).

**Figure 8 nanomaterials-09-01516-f008:**
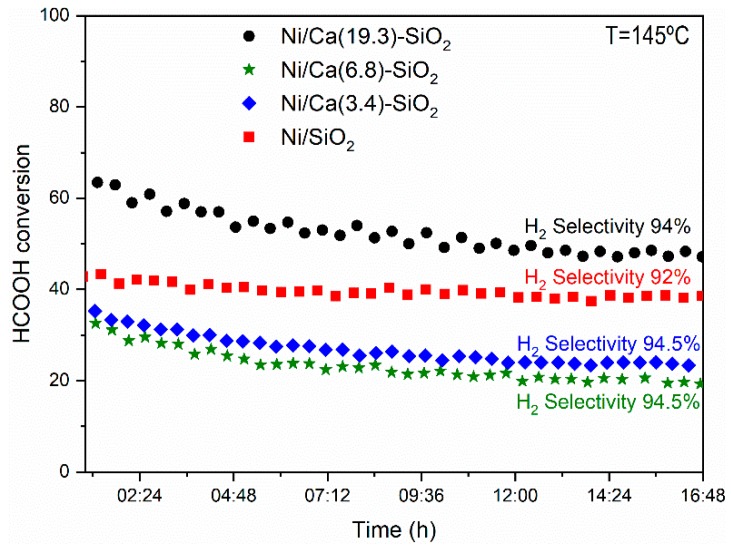
Stability test of the catalysts after reduction at 400 °C in the fixed-bed reactor. (Reaction temperature = 145 °C, W/F = 5 × 10^−5^ g·h·mL^−1^, feed composition: 6% HCOOH/N_2_.)

**Figure 9 nanomaterials-09-01516-f009:**
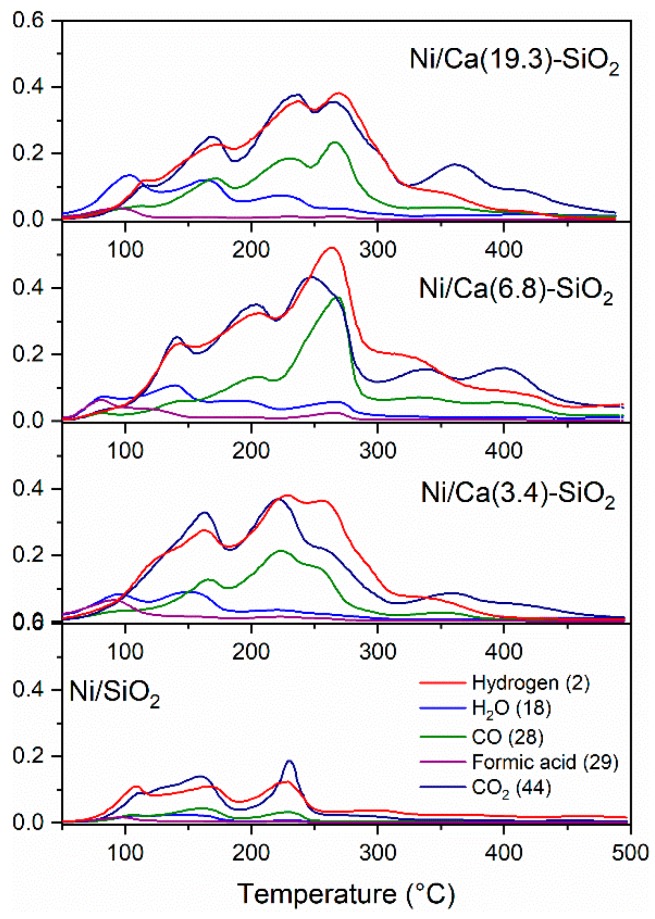
Temperature-programmed surface-reaction profiles of Ni catalysts after 40 Torr pulse of HCOOH at 40 °C.

**Figure 10 nanomaterials-09-01516-f010:**
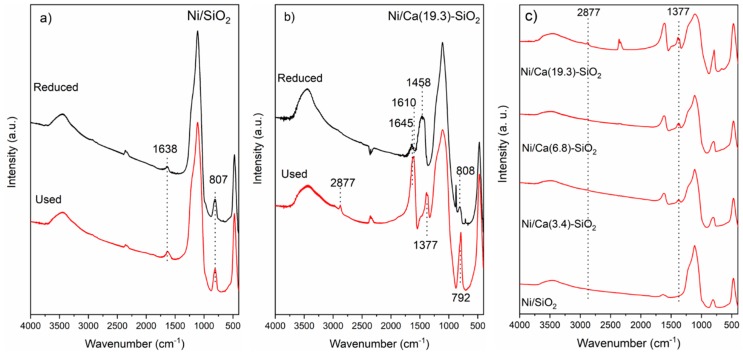
Fourier transform infrared spectroscopy (FTIR) spectra of (**a**) reduced and used Ni/SiO_2_; (**b**) reduced and used Ni/Ca(19.3)-SiO_2_; and (**c**) used Ni/Ca(X)-SiO_2_ catalysts.

**Figure 11 nanomaterials-09-01516-f011:**
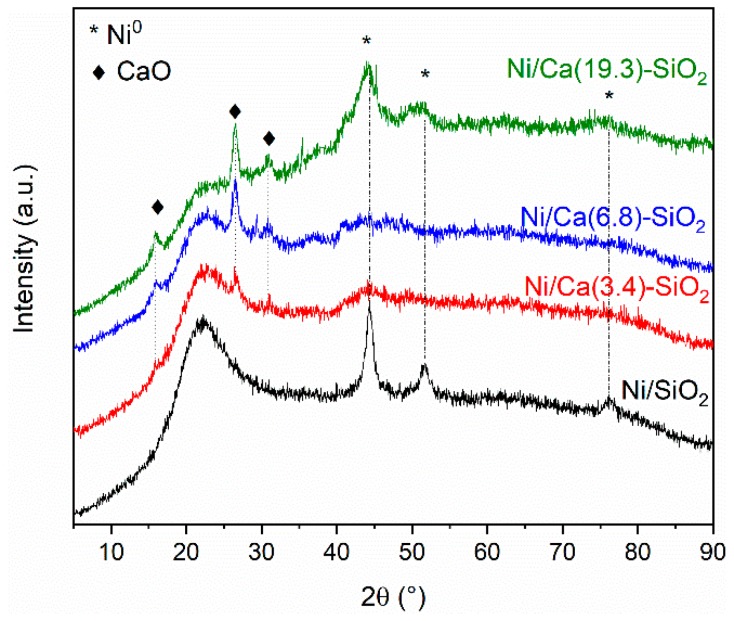
X-ray diffractograms of used Ni/SiO_2_ and Ni/Ca(X)-SiO_2_ catalysts.

**Figure 12 nanomaterials-09-01516-f012:**
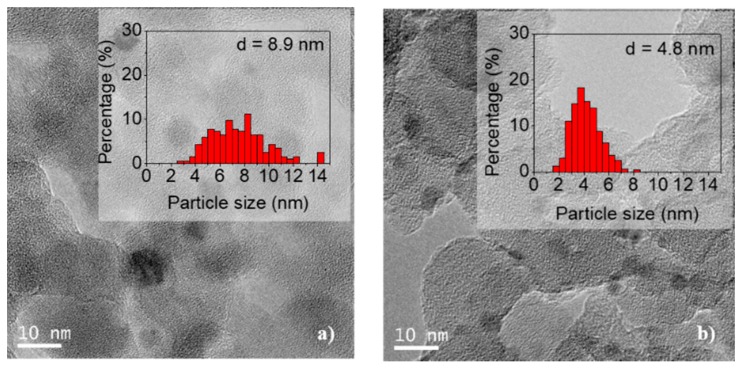
TEM images of used samples: (**a**) Ni/SiO_2_ and (**b**) Ni/Ca(19.3)-SiO_2_; the histograms were included.

**Figure 13 nanomaterials-09-01516-f013:**
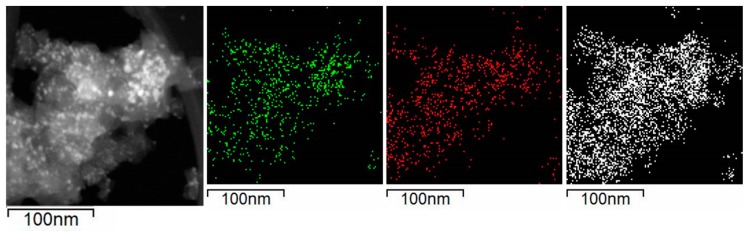
Selected area for the EDX mapping for used Ni/Ca(19.3)-SiO_2_ catalyst; mapping of nickel (green), calcium (red), and silicon (white).

**Table 1 nanomaterials-09-01516-t001:** Catalytic activity of Ni solids: temperature reaction and H_2_ selectivity for 50% and 100% of HCOOH conversions (W/F = 5 × 10^−5^ g·h·mL^−1^; feed composition: 6% HCOOH/N_2_).

Catalyst	Ca/Ni Ratio	T_50%_	S_50%_	T_100%_	S_100%_	H_2_/CO_2_ Ratio
Ni/SiO_2_	-	148	91	180	87	1.01
Ni/Ca(3.4)-SiO_2_	1	153	93	180	91	1.08
Ni/Ca(6.8)-SiO_2_	2	153	93	180	90	1.04
Ni/Ca(19.3)-SiO_2_	5.6	145	92	160	92	0.95

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
