# Peer review of "Hydrogen Production by Formic Acid Decomposition over Ca Promoted Ni/SiO2 Catalysts: Effect of the Calcium Content"

_nanomaterials, 2019, doi:10.3390/nano9111516_

Round 1

Reviewer 1 Report

Paper entitled “ Green hydrogen production by formic acid decomposition over Ca promoted Ni/SiO2 catalysts: effect of the calcium content” represents an original research paper devoted to study Ni catalysts supported on a SiO2 matrix in the decomposition reaction of formic acid in the vapor phase. The effect of doping the support with different loadings of calcium is studied. Paper describes well prepared experimental study including the characterization of catalysts by X-ray diffraction (XRD), X-ray photoelectron spectroscopy (XPS), temperature-programmed reduction (TPR), Fourier transform infrared spectroscopy (FTIR), transmission electron microscopy (TEM) and programmed temperature surface reaction (TPSR). Generally speaking, a rather large experimental work was carried out using standard (for this research area) methods. New catalysts were proposed. Paper is rather well written and structured. The production of H2 from formic acid using heterogeneous catalysts has been previously studied in liquid  and vapor phases, but in most cases formulations based on noble metals, such as Rh, Pt, Ru, 48 Au, Ag, and Pd supported on C, Al2O3 and SiO2 have been investigated.

In my opinion, the presented paper is worth to be published.

Recommendations:

Delete green from the title Where are SEM images of catalysts?

Author Response

Thanks for your remarks. We have considered them and accordingly:

The word “Green” has been deleted from the title. Sorry but we have not performed SEM measurements on our catalysts. However, we have studied our reduced and used catalysts in a HRTEM. We show in the manuscript TEM images (Figures 5 and 12) and HAADF images in STEM mode together with the elemental mapping (Figures 6 and 13).

Reviewer 2 Report

Nice piece of work, the only addition I would ask for is a bit more of an explanation as to why sample Ni/Ca(19.3)-SiO2 shows so much higher stability than the other Ca doped materials in the conclusions

Author Response

Thanks for your remark. To account for your query, we have added in the conclusion section the paragraph: “The doping with Ca modified the interaction of the metal with the support and the Ni particles remained stable during the catalytic test. However, for Ni/SiO2 catalyst the distribution and average particle size are doubled during reaction with respect to those of the reduced sample. The stability in the average size of the nickel particles in the used Ni/Ca(19.3)-SiO2 catalyst could explain the difference in the behavior of this material to the others. Moreover, this catalyst was relatively stable under the reaction conditions used presenting an equal performance in two sequential catalytic tests.”

Reviewer 3 Report

The work is clearly organized, the experiments are sufficiently described and the discussion is logical.

Not even minor problems were encountered.

Publication in this form is recommended

Author Response

Thanks for your remark.

Reviewer 4 Report

In this study, the application of Ni/SiO2 catalysts with different calcium content as green hydrogen production. The author report a serics of Ni/SiO2 heterogeneous catalysts with different calcium content in the catalytic dehydrogenation of FA to generate hydrogen. Overall the work is of well researched and written into a nice paper with helpful diagrams. I think the paper will be of interest to the readership of nanomaterials and I recommend the paper is accepted with some minor changes.

Ni catalysts with different calcium content are the same SBET? If not, the activity are increasing or decreasing? Can the material be reused? In the reaction, why does the calcium content increase and the activity increase?

Author Response

Thanks for your remarks:

The specific surface area (SBET) decreases when SiO2 (200m2/g) is impregnated with Ca salt. So, for the Ca(19.3)-SiO2 support, with the highest CaO loading, the surface area becomes 73 m2/g (see reference 24 in the manuscript). This can be explained considering that the binary support undergoes structural changes with the addition of CaO. Consequently, the activity of the catalyst increases with the Ca content even if the specific surface area decreases. This catalytic behavior is due to the improved interaction of Ni particles with the Ca doped support stabilizing these Ni nanoparticles against sintering in reaction. A paragraph has been added in the conclusion section to account for this point. Ca(19.3)-SiO2 catalyst was stable under the reaction conditions used presenting an equal performance in two sequential catalytic tests.